# Meaning in Life among Older Adults: An Integrative Model

**DOI:** 10.3390/ijerph192416762

**Published:** 2022-12-14

**Authors:** Lee Greenblatt-Kimron, Maya Kagan, Ester Zychlinski

**Affiliations:** School of Social Work, Ariel University, Ariel 40700, Israel

**Keywords:** integrative model, meaning in life, older adults, personality characteristics, psycho-social factors

## Abstract

Meaning in life (MIL) among older adults has a significant physical and mental health impact. This study aimed to present an integrative model of factors that contribute to variability in MIL among older adults, including background characteristics (gender, age, employment status, religiosity), personality characteristics (locus of control, self-efficacy, optimism), and psycho-social factors (psychological distress and loneliness). Participants (751 older adults, *M_age_* = 72.27, *SD* = 6.28; 446 female, 305 male) responded to a questionnaire in-person or online. Measures included: demographic variables, Short Scale for the Assessment of Locus of Control, New General Self-Efficacy Scale, Life Orientation Test–Revised, Kessler Psychological Distress Scale, and Hughes Short Scale for Measuring Loneliness. Hierarchical regression revealed that younger and religious older adults reported higher MIL levels than older and non-religious older adults. Internal locus of control, higher self-efficacy, and higher optimism were linked to higher MIL levels. Higher psychological distress and loneliness were associated with lower MIL levels, with psychological distress contributing the most of all variables in the study model to explain the variance in MIL among older adults. Employed older old adults reported lower MIL levels than those unemployed. The study emphasizes the importance of an integrative approach in the examination of MIL among older adults.

## 1. Introduction

Meaning in life (MIL) refers to a person’s sense of understanding, purpose, and significance [1,2]. Frankl [3] contended that people’s fundamental motive is to create MIL. MIL is defined as people’s perception of themselves and the world, understanding their place in the world, and realizing what they are striving to accomplish [4]. MIL is presumed to be cognitive in nature and usually formed early in life; yet, can be changed by personal experiences [2]. Researchers have shown links between higher MIL with positive emotions including feelings of love, happiness, and vibrancy. Conversely, MIL is inversely correlated with negative emotions including fear, anger, embarrassment, and sorrow [5].

Meaning in life is regarded as a personal experience [6] linked with psychological well-being throughout the lifespan [7]. Nevertheless, MIL varies during the lifespan, with some authors suggesting that life events and aging change a person’s understanding of MIL [8], while others maintain that older age heightens one’s sense of MIL [6]. Although meaning in life is essential in all stages of life, the psychosocial development theory emphasizes the importance of MIL in old age [9], postulating that in the eighth stage older adults are faced with the crisis of integrity vs. despair regarding whether their lives were meaningful. Today, the final age stage is much longer and is divided into periods on a continuum from young older people (65+) the so-called “baby boomers” to older older people (80+) the “silent generation” [10]. Coping with the crisis of integrity vs. despair may change throughout the years of old age and may affect MIL in this population. Indeed, extensive research has underscored the significance of MIL in old age by linking MIL with diverse manifestations of successful aging [11] including better physical health, lower mortality [12], slower cognitive decline [13], higher subjective well-being [11], positive affect [13], fewer depressive symptoms [13], and better psychological adjustment [14]. Moreover, research suggests that MIL after traumatic experiences promotes post-traumatic growth [15], including older adults who survived early life trauma [16]. In an effort to explore factors contributing to variability in MIL in general, and particularly among older adults, previous studies explored single variables including age [17], culture [18], human relationships [19] and engagement in activities [20]. Nevertheless, conceptual models integrating diverse variables are needed in order to broaden the understanding of MIL [1]. In this light, fostering older adults’ well-being requires a more profound knowledge of the factors that promote a sense of MIL in older age [21]. Therefore, the current research focuses on possible differences that may apply to this coping on their MIL. Identifying the factors that promote or diminish MIL in old age could help to develop appropriate therapeutic intervention methods and even adjust public policy. Specifically, the current study aims to present an integrative model of factors contributing to MIL among older adults, namely background characteristics (gender, age, employment status, religiosity), personality characteristics (locus of control, self-efficacy, optimism), and psycho-social factors (psychological distress and loneliness).

### 1.1. Background Characteristics and Meaning in Life

MIL may relate to background characteristics, demonstrated in a study examining differences in age, gender, and ethnicity in relation to MIL. Findings show that younger participants (40 and below) perceive materialistic concerns as more significant to MIL than older participants (60 and above). In contrast, older participants reported family and communal values as more essential to MIL than younger participants [18]. Significant gender differences in MIL have been reported, with women reporting higher levels of presence and searching for MIL than men across the life span [4]. Lower MIL was reported in males than females among community-dwelling older adults in a cross-sectional survey in four European countries [22]. Nevertheless, contradictory findings have been reported regarding the link between MIL and gender among older adults. While in one study almost no gender differences were found among older adults [18], a meta-analysis examining MIL at advanced ages showed lower MIL among older women than men [13].

Regarding employment, the socio-emotional selectivity theory [23] suggests that older adults strive to fulfill emotional needs through work and deeper interpersonal co-worker relationships. Studies have shown that meaningful work is vital for older workers [4] demonstrated by higher motivation, increased significant relationships, and overall well-being [24]. Conversely, loss of employment due to retirement may have adverse effects on MIL [24].

Although MIL is not fundamentally religious in nature [25], studies show a link between religion with MIL [26], particularly as people age [27]. Religion offers beliefs that assist making sense of life’s encounters, including disaster and distress [28]. Moreover, spirituality invigorates purpose by clarifying goals, including reinforcing a relationship with the divine, attaining salvation, and living a righteous life [29].

### 1.2. Personality Characteristics and Meaning in Life

Previous studies noted that personality characteristics were essential variables for MIL [30,31], including a negative correlation with neuroticism [32], and positive correlations with extroversion, conscientiousness, and agreeableness [30]. Important to the current study, links have been reported between MIL and locus of control, self-concept (i.e., self-efficacy) [33], and optimism [30].

### 1.3. Locus of Control

Locus of control refers to the belief that events in life are controlled internally or externally [34]. People with an internal locus of control perceive self-accountability for shaping and controlling their lives, while those with an external locus of control perceive being controlled by luck, fate, opportunities, others, and events [35]. A high internal locus of control was linked with physical and mental outcomes including lower pain [36], better physical functionality [36], higher self-reported health [37], lower mortality [38], and better quality of life [39]. Meaning in life is based on feelings of coherence, purpose and existential mattering [40], although both of these concepts relate to feelings of coherence and purpose. However, there is a lacuna in studies examining the link between locus of control and MIL.

### 1.4. Self-Efficacy

According to the social-cognitive theory [41], self-efficacy is the belief in one’s capability to perform activities that allow for deliberate outcomes. This belief plays a central role in emotional self-regulation, self-enablement vs. self-debilitation, and perceptions of successes vs. failures [41]. Therefore, people with low self-efficacy tend to internalize failure, with ensuing feelings of depression and helplessness [41]. Correlations between self-efficacy and physical and mental health have been identified, including better quality of life [42], and life satisfaction [43], while negative correlations have been shown with psychological distress [44], anxiety [43], and depression [45]. Reduced self-efficacy among older people in social situations may result in age-related physical changes, loss of previous social roles, and decreased formation of new social relationships [46]. A link was also found between parental self-efficacy and loneliness among older adults [47]. An association was shown between self-efficacy and MIL in the general Norwegian population [48]; however, to the best of our knowledge, the contribution of self-efficacy to explaining variability in MIL among older adults has yet to be examined.

### 1.5. Optimism

Optimism reflects positive anticipation of future outcomes [49]. Dispositional optimism is conceived as two disparate metrics, optimism and pessimism, together comprising a measure of prospects for positive outcomes from future events [50]. Optimistic people tend to persevere when faced with challenges during goal-seeking, while pessimistic individuals tend to retreat from their efforts [51]. Optimism’s positive bias appears to be a mechanism for evaluating one’s life [52], which, in line with the psychosocial development theory [9], is critical in one’s older years. Indeed, studies show correlations between higher optimism in older adults and better self-reported health [53], greater positive affect [54], positive emotions [50], and life satisfaction [52]. Numerous studies among the general population have shown a correlation between positive outlooks such as optimism and high MIL levels [30], nevertheless, there is a need to examine this association among older adults.

### 1.6. Psycho-Social Factors and Meaning in Life

#### 1.6.1. Psychological Distress

Psychological distress refers to a mental health concern consisting of depression, anxiety, and somatic symptoms disturbing a person’s functioning [55]. Mixed findings are reported of associations between age and psychological distress. Some studies show higher psychological distress among younger than older adults [56], while others show higher rates of mild to moderate psychological distress among older than younger adults [57]. Psychological distress has been linked with an increased risk of all-cause mortality [58]. Important to the current study, associations have been found between successful coping with psychological distress and finding MIL [3], and between less psychological distress and lower psychopathology with enhanced MIL [1]. Nevertheless, there are insufficient studies of psychological distress among older adults [59].

#### 1.6.2. Loneliness

Loneliness comprises perception of unsatisfied personal, emotional, and social needs [60], being defined as the difference between aspired and existing social relations [61]. At advanced ages, contributing factors to loneliness include loss or absence of relationships with others [62] and insufficient quality relationships [63]. Later years create conditions whereby older people exhibit more loneliness than younger counterparts [64], with studies indicating about one-third of older people experiencing loneliness [65], and those aged 80 and over regularly feeling lonely [60]. It was also found that older men with the personality characteristics of agreeableness, emotional stability, and openness reported feeling less lonely [66]. Specifically, research links loneliness with reduced MIL [67]; both are essential for daily functioning and maintaining mental health and well-being in older age [68].

### 1.7. Research Hypotheses

Demographic factors: Being female, younger, employed, and religious will be associated with higher MIL levels than being male, older, unemployed and not religious.

Personality characteristics: An internal locus of control, higher self-efficacy levels, and higher optimism levels will be associated with higher MIL levels.

Psycho-social factors: Higher psychological distress and loneliness levels will be associated with lower MIL levels.

## 2. Materials and Methods

### 2.1. Sampling Method and Data Collection

The study was approved by the Ariel University Ethics Committee (approval number “AU-SOC-EZ-20191029”). Written explanation of the study was provided and signed informed consent was obtained before participation. Research assistants distributed structured questionnaires to a convenience sample, mostly in-person (e.g., relatives and friends, nursing homes, and assisted living housing). Some questionnaires were completed online via social networks for older populations. The response rate for in-person recruitment was about 85%; the response rate for questionnaires distributed online cannot be estimated since it is unknown how many people were exposed to them without responding. Informed consent was obtained from all subjects involved in the study.

### 2.2. Research Instruments

#### 2.2.1. Independent Variables

Demographic variables included: gender (0—male, 1—female), age (in years), religiosity (0—religious, 1—not religious) and employment status (0—employed, 1—unemployed).

Locus of control was assessed by the Short Scale for the Assessment of Locus of Control (IE-4) designed by Kovaleva [69], including two items on the Internal Locus of Control scale and two items on the External Locus of Control scale. Responses ranged from 1 (doesn’t apply at all) to 5 (applies completely). After reverse scorning of the External Locus of Control scale, an index was created from calculating the average of responses to all items; higher scores indicate higher levels of internal locus of control while lower scores indicate higher levels of external locus of control. The current study’s internal consistency reliability (Cronbach’s alpha) was 0.603.

Self-efficacy was assessed by the New General Self-Efficacy Scale (NGES) [70] consisting of 8 questions, each having 3 responses: 1 (not at all), 2 (moderately), and 3 (to a high degree). A total score was calculated by summing all eight questions (i.e., from 8 to 24), with higher scores indicating higher self-efficacy. The current study’s internal consistency reliability (Cronbach’s alpha) was 0.864.

Optimism was assessed by the 10-item Life Orientation Test–Revised (LOT-R) [71], comprised of 3 items measuring optimism, 3 items measuring pessimism, and 4 items serving as fillers. Items are rated on a four-point Likert-type scale, ranging from 0 (strongly disagree) to 4 (strongly agree). Items related to pessimism were revised. Excluding filler responses, total scores are calculated by summing the remaining six items (resulting in scores ranging from 0 to 24) with higher scores indicating higher optimism. The current study’s internal consistency reliability (Cronbach’s alpha) was 0.672.

Psychological distress was assessed by the six-item Kessler Psychological Distress Scale (K6) [72]. It examines nervousness, hopelessness, irritability, negative affect, fatigue, and worthlessness, experienced over the past 30 days. Items are rated on a five-point Likert scale (revised) from 0 (absence of the symptom) to 4 (highest level of the symptom). Total scores range from 0 to 24, with higher scores indicating higher psychological distress. The current study’s internal consistency reliability (Cronbach’s alpha) was 0.906.

Self-reported loneliness was assessed by a short scale for measuring loneliness comprising of a three-item scale [73]. Respondents were asked how often they feel they lack companionship, feel left out, and feel isolated from others. Response ranged from 1 (hardly ever), 2 (some of the time), or 3 (often). Total scores range from 3 to 9, with higher scores indicating higher loneliness. The current study’s internal consistency reliability (Cronbach’s alpha) was 0.898.

#### 2.2.2. Dependent Variable

Presence of meaning in life. The Meaning in Life Questionnaire (revised) [5] consists of two five-item factors designed to measure presence of meaning (to what degree respondents feel that their lives have meaning), and search for meaning (to what degree respondents strive to find meaning and understanding in their lives). The present study referred only to the presence of meaning in life subscale, which included items such as “I understand my life’s meaning”, “I have a good sense of what makes my life meaningful”. Items were rated on a 7-point Likert-type scale ranging from 1 (absolutely untrue) to 7 (absolutely true). The final index was calculated as the sum of responses to the five items, with scores ranging from 7 to 35 and higher scores indicating a higher level of presence of MIL. The current study’s internal consistency reliability (Cronbach’s alpha) was 0.864. For descriptive statistics of the research variables see Table 1.

## 3. Results

### Characterization of the Sample

Community dwelling Israeli adults aged 65 and older participated in the study (*N* = 751; 59.4% women). Their age ranged from 65 to 85 years (M = 72.27, SD = 6.28) and their formal education level ranged from 4 to 30 years (M = 13.78, SD = 3.35). Of the respondents, 59.4% were religious, 61.7% were unemployed, 65.9% were married or in a committed relationship, 21.2% widowed, 10.4% divorced, and 2.5% single. Most respondents (54.3%) classified themselves as having a medium socioeconomic status, 38.1% as having a high status, and 7.6% as having a low status. See Table 1 for the characterization of the sample and distribution of the research variables.

A hierarchical regression analysis examined the association between selected demographic, personality, and psycho-social factors, and MIL among older Israeli adults (see Table 2). The maximal VIF of predictors was 1.77, indicating the assumption of multicollinearity in the regression model was rejected. Gender, age, employment status, and religiosity were entered in the first step of the regression (*F*_(4,746)_ = 10.26, *p* < 0.001). Locus of control, self-efficacy, and optimism were entered in the second step (*F*_(7,743)_ = 69.07, *p* < 0.001) and psychological distress and loneliness were entered in the third step (*F*_(9,741)_ = 73.07, *p* < 0.001). Together, the independent variables accounted for 46.4% of the variance in MIL in the sample.

The final (third) step of the regression model revealed that those who were younger and religious reported higher MIL than those older (β = −0.058, *p* < 0.05) and not religious (β = −0.084, *p* < 0.01). Having a higher internal locus of control (β = 0.132, *p* < 0.001), higher self-efficacy (β = 0.146, *p* < 0.001), and higher optimism (β = 0.225, *p* < 0.001) were associated with higher MIL. In contrast, higher psychological distress (β = −0.235, *p* < 0.001) and loneliness (β = −0.157, *p* < 0.001) were associated with lower MIL. Notably, employed older adults reported lower MIL than those who were unemployed (β = 0.100, *p* < 0.001). No association was found between gender and MIL (*p* > 0.05).

## 4. Discussion

To the best of our knowledge, this is the first study examining an integrative model of factors contributing to MIL among older adults, including background characteristics, personality characteristics, and psycho-social factors. Most of the hypotheses were confirmed except two background characteristics: gender and employment.

The study’s findings reveal that younger participants (younger older adults) reported higher MIL levels than older participants (older older). This may be explained by losses associated with aging, including degenerating physical and mental health and decreasing intimate relationships [74]. Although losses characterize old age, the findings show that there are probably differences in the intensity of the losses manifested throughout the entire aging period. Moreover, among the oldest age group (above 85 years) adverse life experiences may reduce a sense of coherence, a vital aspect of MIL [75]. The current results are in line with the findings of an integrative literature review suggesting that maintaining MIL at older ages may be challenging [19]; in particular for non-religious older adults, therefore, it is essential to develop and implement intervention plans. On a clinical level, logotherapy, a type of psychotherapy based on the assumption that the search for meaning, even at times of suffering, includes a possible resolution to human suffering [76] may help older adults find personal MIL. Facilitating a combination of individual and community interventions may be essential to enriching MIL among older adults, such as promoting community contact and engagement including intergenerational volunteer-based neighborhood programs.

Religious participants reported more MIL than non-religious participants, supporting previous studies indicating that it is common for people to gain MIL from religion and spirituality, especially as they age [27]. Our findings also support the notion that those who possess religiously based perceptions of the world (e.g., regarding death) have a more profound sense of MIL [28]. Based on the current findings, together with the findings of both qualitative studies [27] and quantitative studies [26] from a policy planning perspective, it may be useful to promote social programs within existing religious systems so that the contribution of religion to MIL can be further strengthened.

Gender was not a predictor of MIL in our findings, with similar levels of MIL reported in both genders. This partially supports a previous study [18] suggesting a stronger association between MIL and cultural and age differences than gender differences. Nevertheless, the current findings contradict previous research showing gender differences in MIL throughout the course of life and into old age [4], as well as lower MIL in older males than females [22]. On a practical level, this finding provides evidence for the need of individual and communal interventions for both genders.

An important finding in the current study is lower MIL among employed older adults than those unemployed. This unexpected result is in contrast to previous studies [24] and may have vital implications for individual, community, and legislative interventions with older adults. Perhaps continued employment is due to financial need rather than a source of meaning. In Israel, only 46% of older adults have a pension; therefore, the majority are forced to depend on state financial benefits, which are insufficient for standard living expenses of older adults [77]. Moreover, a previous study in Israel reported that many older Israeli parents provide financial assistance to their adult children [78]; therefore, older adults may feel a burden, or, a loss of purpose if relying on their adult children for financial assistance. Moreover, meaningful work is essential for older workers [4], supporting the socio-emotional selectivity theory [23] suggesting that older adults aspire to satisfy their emotional needs through work, with a particular need for deeper interpersonal relations with co-workers. Hence, the finding in the present study raises questions regarding the relationship between MIL and employment. Specifically, the psychological outcomes of employment vs. retirement among older adults, and the degree to which they perceive their work as meaningful. Future studies should examine differences in MIL between older employed adults who work as a means to financially survive compared to those you choose to work regardless their financial status. Future studies should also examine the effect of earning more compared to earning less in the association between employment and MIL among older adults. On an individual level, practitioners should focus on employment status and its impact on MIL. This may allow for a deeper understanding of the overall well-being of older adults and bring insight into the dynamics of intergeneration family relations with regard to which generation is supporting which. On the legislative level, the current study findings suggest that increasing state allowances for older adults is recommended, as well as increased societal awareness of the interplay between their financial and psychological well-being. On a policy level, it is crucial to consider the level of benefits older adults receive as well as laws that protect their rights in the workplace.

In terms of personality characteristics, the current findings indicate that internal locus of control is associated with higher MIL. This supports the limited research examining this association among teachers [33], suggesting that when one’s personal resources are internal, it may be easier to benefit and derive meaning, rather than being dependent on a particular constellation of external circumstances or motivation.

Similarly, higher self-efficacy among the participants of the present study was found to be associated with higher MIL, supporting studies showing a link between self-efficacy in coping and improved quality of life [42]. Moreover, coinciding with findings of a link between self-efficacy and MIL among the general Norwegian population [48], the present findings highlight the contribution of self-efficacy to MIL in older adults, indicating that better coping and achieving skills (salient aspects of self-efficacy) may create a greater sense of MIL as people age.

Findings show a correlation between higher optimism with higher MIL. This finding supports various studies [30] reporting a relationship between optimism and MIL. This may be particularly relevant for older adults, as optimism is a known as a mechanism for appraising one’s life [52], which according to the psychosocial development theory is critical at advanced ages [9]. One way to strengthen optimism, locus of control and self-efficacy is to offer workshops allowing older adults to acquire new skills that decrease dependence on their surroundings and encourage new possibilities. For example, hub centers designed to promote skills, employment, and volunteering for the older adults that also provide personal counseling and intergenerational programs with mutual activities would likely increase their sense of self-efficacy and enhance MIL.

The present findings highlight a negative link between psychological distress and MIL. Psychological distress contributed the most of all variables in the study model to explain the variance in MIL among this cohort, which is central to the study of older adults as few studies have focused on psychological distress among older adults [59]. The current results suggest that psychologically distressed older adults have less MIL, which may hinder a positive resolution to Erikson’s [9] final developmental stage of integrity vs. despair.

Furthermore, a negative relationship was found between loneliness and MIL, supporting previous research [67]. Loneliness is known as one of the most destructive phenomena in old age, stemming from many sources including physical, cognitive, social, economic, and technological limitations that prevent participation in social activities with maintaining and creating social relationships. Accordingly, life may become less interesting, satisfying, and meaningful. Interventions should aim to broaden older adults’ social circles and sense of belonging. Moreover, national policies should promote successful aging strategies to prevent loneliness and psychological distress and thereby increase MIL.

The present study has some limitations. First, it used a cross-sectional design; therefore, cannot provide causal explanations. Second, about 15% of the data were collected through online sampling, so the results may be biased towards those with computer literacy and accessibility [79]. Additionally, the internal consistency reliability of the optimism and locus of control scales in the current study is low, while in contrast to previous studies optimism had a strong internal reliability of 0.82 [80] and internal and external locus of control had an internal reliability of 0.69 and 0.71, respectively in a Norwegian sample and 0.80 and 0.60, respectively in a German-speaking sample [81]. Therefore, future studies should examine whether these scales are indeed suitable for older people and whether they are sensitive to the cultural aspect of the population in which they were administered. In future studies the current research model should also be explored using alternative research tools. Finally, the study addressed specific factors. Future studies should examine combinations of additional variables such as the impact of having a pension, caring for grandchildren, and mental disorders (e.g., post-traumatic stress disorder). Despite these limitations, the study model broadens the understanding of MIL among older adults, with vital practical implications for individual, community, and legislative interventions in old age.

## 5. Conclusions

It has been suggested there is a need for conceptual models and theoretical frameworks that integrate findings to broaden the understanding of the nature of MIL [1]. As such, one of the current study’s main strengths is it is the first to use an integrative model incorporating diverse variables to explore their contribution to variability in MIL in general, and specifically among older adults. We suggest that future studies replicate a similar model to compare older adults’ behavior with that of their younger counterparts. Moreover, the knowledge gained by this unique study model has potentially significant practical implications. First, it supports more tailored treatment approaches for older people by predicting demographic and personality characteristics and psycho-social factors that may contribute to MIL. As the current study shows no relationship between gender and MIL among older adults, promotion of MIL among both genders should be equivalent. Moreover, employment did not contribute to MIL in the current sample of older adults. This result provides important clinical implications as well as directions for future studies, underscoring the possibility that employment in not necessarily linked with MIL in older age. Further, as the study findings indicate older adults experiencing psychological distress appear to be at higher risk of experiencing a reduced sense of MIL, this underscores the need for clinicians to focus on helping older adults find a sense of MIL. Moreover, it is recommended that community interventions be implemented as part of an overall national policy designed to enhance older people’s sense of belonging and MIL, as described in the examples above. Interventions that increase older adults’ internal loci of control, self-efficacy, and positive outlooks (e.g., optimism) should be implemented with the aim of enhancing their perception of competence and self-esteem and increasing their sense of MIL and resilience. Likewise, interventions should aim to broaden older adults’ social circles and sense of belonging, to address the loneliness found to have a strong link with MIL in old age.

## Figures and Tables

**Table 1 ijerph-19-16762-t001:** Distribution of the research variables (*n* = 751).

Variables		*N* (%)	Mean	SD
Gender	Male	305 (40.6)		
	Female	446 (59.4)		
Age			72.27	6.28
Employment status	Employed	289 (38.5)		
	Not employed	462 (61.5)		
Religiosity	Religious	446 (59.4)		
	Not religious	305 (40.6)		
Locus of control			3.71	0.69
Self-efficacy			17.22	3.91
Optimism			21.78	4.41
Psychological distress			11.81	5.18
Loneliness			4.55	1.74
Presence of meaning in life			25.89	6.79

**Table 2 ijerph-19-16762-t002:** Summary of the hierarchical regression analysis for variables predicting the self-reported presence of meaning in life (*n* = 751).

		B	Std. Error	Beta	T	Adj. R^2^	ΔR^2^	F
Step 1						0.047	0.052	*F*_(4,746)_ = 10.26 ***
	Gender	−0.750	0.49	−0.054	−1.52			
	Age	−0.220	0.04	−0.204	−5.44 ***			
	Employment status	1.635	0.52	0.117	3.14 **			
	Religiosity	−1.635	0.50	−0.118	−3.30 ***			
Step 2						0.388	0.342	*F*_(7,743)_ = 69.07 ***
	Gender	−0.058	0.40	−0.004	−0.14			
	Age	−0.096	0.03	−0.089	−2.90 **			
	Employment status	1.379	0.42	0.099	3.30 ***			
	Religiosity	−1.127	0.40	−0.082	−2.80 **			
	Locus of control	1.800	0.33	0.184	5.42 ***			
	Self-efficacy	0.376	0.06	0.216	6.42 ***			
	Optimism	0.548	0.05	0.356	10.89 ***			
Step 3						0.464	0.076	*F*_(9,741)_ = 73.07 ***
	Gender	0.203	0.37	0.015	0.54			
	Age	−0.062	0.03	−0.058	−2.00 *			
	Employment status	1.400	0.39	0.100	3.57 ***			
	Religiosity	−1.158	0.38	−0.084	−3.07 **			
	Locus of control	1.288	0.32	0.132	4.08 ***			
	Self-efficacy	0.254	0.06	0.146	4.53 ***			
	Optimism	0.345	0.05	0.225	6.77 ***			
	Psychological distress	−0.308	0.05	−0.235	−6.59 ***			
	Loneliness	−0.613	0.13	−0.157	−4.70 ***			

*p* < 0.05 *, *p* < 0.01 **, *p* < 0.001 ***.

## Data Availability

Not applicable.

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
