# Peer review of "Meaning in Life among Older Adults: An Integrative Model"

_ijerph, 2022, doi:10.3390/ijerph192416762_

Round 1
Reviewer 1 Report
This paper is flat and stable, lacking innovation and cannot attract a large number of readers, because the paper is easy to understand and cannot be called excellent. It is suggested that the authors can present the results brilliantly, but some are disappointed that they are too simplistic. Another item, the data in the table and the text data should be consistent, and the data should not be wrong. For example Line221.
Author Response
Reviewer #4: This paper is flat and stable, lacking innovation and cannot attract a large number of readers, because the paper is easy to understand and cannot be called excellent. It is suggested that the authors can present the results brilliantly, but some are disappointed that they are too simplistic. Another item, the data in the table and the text data should be consistent, and the data should not be wrong. For example Line221.
Answer: We thank Reviewer #4 for the feedback. As evident from our literature review, MIL is a complex variable that might be explained by an extensive range of factors. Our study, similar to most studies on MIL and in other domains, does not presume to reach a full explanation of the dependent variable. It is doubtful whether this is possible on a statistical or theoretical level, considering that there are also latent variables of which we may lack knowledge. At the same time, we are aware that additional variables might explain MIL, and in this study, we propose additional research variables that should be considered in future studies. The current study joins previous research and contributes to the existing knowledge on MIL among older people. The findings show that, besides the significance of the psychosocial variables in explaining MIL, demographic variables might help identify older adults whose MIL is lower than others and provide them with proper attention and assistance. Moreover, the study reveals that individuals' personality traits significantly affect their perception of life as meaningful. This finding might help develop new directions for intervening and treating older people, as it stresses the connection between their al mental resources and their perception of meaning in life.
We also revised the paper with the aim of writing it in a more concise manner and added a broader explanation of the purpose of the study in the Introduction section:
Meaning in life is regarded as a personal experience [6] linked with psychological well-being throughout the lifespan [7]. Nevertheless, MIL varies during the lifespan, with some authors suggesting that life events and aging change a person's understanding of MIL [8], while others maintain that older age heightens one's sense of MIL [6]. Although meaning in life is essential in all stages of life, the psychosocial development theory emphasizes the importance of MIL in old age was highlighted by the psychosocial development theory [9], postulating that in the eighth stage older adults are faced with the crisis of integrity vs. despair regarding whether their lives were meaningful. Today, the final age stage is much longer and is divided into periods on a continuum from young older people (65+) the so-called "baby boomers" to older older people (80+) the "silent generation" [10]. Coping with the crisis of integrity vs. despair may change throughout the years of old age and may affect MIL in this population. Indeed, extensive research has underscored the significance of MIL in old age by linking MIL with diverse manifestations of successful aging [11] including better physical health, lower mortality [12], slower cognitive decline [13], higher subjective well-being [11], positive affect [13], fewer depressive symptoms [13], and better psychological adjustment [14]. Moreover, research suggests that MIL after traumatic experiences promotes post-traumatic growth [15], including older adults who survived early life trauma [16]. In an effort to explore factors contributing to variability in MIL in general, and particularly among older adults, previous studies explored single variables including age [17], culture [18], human relationships [19]and engagement in activities [20]. Nevertheless, conceptual models integrating diverse variables are needed in order to broaden the understanding of MIL [1]. In this light, fostering older adults' well-being requires a more profound knowledge of the factors that promote a sense of MIL in older age [21]. Therefore, the current research focuses on possible differences that may apply to this coping on their MIL. Identifying the factors that promote or diminish MIL in old age could help to develop appropriate therapeutic intervention methods and even adjust public policy. Hence, Specifically, the current study aims to present an integrative model of factors contributing to MIL among older adults, namely background characteristics (gender, age, employment status, religiosity), personality characteristics (locus of control, self-efficacy, optimism), and psycho-social factors (psychological distress and loneliness).
Added bibliography:
- Schnell, T. Sources of meaning and meaning in life questionnaire (SoMe): relations to demographics and well-being. J. Posit. Psychol. 2009 4, 483–499.
- King, L.A.; Hicks, J.A.; Krull, J.L.; Del Gaiso. A.K. Positive affect and the experience of meaning in life. J Pers Soc Psychol 2006 90(1) ,179-196.
- Reker, G.T.; Peacock, E.J.; Wong, P.T. Meaning and purpose in life and well-being: a life-span perspective. J Gerontol 1987 42, 44–49.
- Lissitsa, S.; Zychlinski, E.; Kagan, M. The Silent Generation vs Baby Boomers: Socio-demographic and psychological predictors of the “gray” digital inequalities. Comput Hum Behav 2022 128, 107098.
- MacKinlay, E. The Spiritual Dimension of Ageing. Jessica Kingsley Publishers. 2017.
Furthermore, we expanded the information of the sample by adding the following information:
Community dwelling Israeli adults aged 65 and older participated in the study (N = 751; 59.4% women). Their age ranged from 65 to 85 years (M = 72.27, SD = 6.28) and their formal education level ranged from 4 to 30 years (M = 13.78, SD = 3.35). Of the respondents, 59.4% were religious, 61.7% were unemployed, 65.9% were married or in a committed relationship, 21.2% widowed, 10.4% divorced, and 2.5% single. Most respondents (54.3%) classified themselves as having a medium socioeconomic status, 38.1% as having a high status, and 7.6% as having a low status.
Additionally, we moved Table 1 for clarity purposes and added the following information about the dependent variable:
Presence of meaning in life. The Meaning in Life Questionnaire (revised) [5] consists of two five-item factors designed to measure presence of meaning (to what degree respondents feel that their lives have meaning), and search for meaning (to what degree respondents strive to find meaning and understanding in their lives). The present study referred only to the presence of meaning in life subscale, which included items such as "I understand my life’s meaning", "I have a good sense of what makes my life meaningful". Items were rated on a 7-point Likert-type scale ranging from 1 (absolutely untrue) to 7 (absolutely true). The final index was calculated as the sum of responses to the five items, with scores ranging from 7 to 35 and higher scores indicating a higher level of presence of MIL. The current study’s internal consistency reliability (Cronbach's alpha) was .864.
Moreover, in order to ensure that the data should not be wrong as pointed out by the Reviewer, we replaced the word "step" instead of the word "model", in order to avoid ambiguity in the reading of the Table.
In addition, in order to clarify that we are referring to older adults at different stages of old age we have made three amendments to clarify this point.
Firstly, in the Introduction section, under the heading Background Characteristics and Meaning in Life we included the age groups that the researchers in previous work referred to, as follows:
MIL may relate to background characteristics, demonstrated in a study examining differences in age, gender, and ethnicity in relation to MIL. Findings show that younger participants (40 and below) perceive materialistic concerns as more significant to MIL than older participants (60 and above). In contrast, older participants reported family and communal values as more essential to MIL than younger participants [18].
Secondly, in the Introduction, we added the following in the section on Loneliness:
Later years create conditions whereby older people exhibit more loneliness than younger counterparts [64], with studies indicating about one-third of older people experiencing loneliness [65], and those aged 80 and over regularly feeling lonely [60]. It was also found that older men with the personality characteristics of agreeableness, emotional stability, and openness reported feeling less lonely [66]. Specifically, Rresearch also links loneliness with reduced MIL [67]; both are essential for daily functioning and maintaining mental health and well-being in older age [68].
Added Reference:
- Itzick, M.; Kagan, M.; Zychlinski, E. The big five personality traits as predictors of loneliness among older men in Israel. J Psychol 2020 154(1), 60-74.
Thirdly, in order to clarify that we referred to older adults at different stages of old age we rewrote the sentence in the Discussion section as follows:
The study’s findings reveal that younger participants (younger older adults) reported higher MIL levels than older participants (older older). This may be explained by losses associated with aging, including degenerating physical and mental health and decreasing intimate relationships [74]. Previous studies have highlighted the effect of age on MIL, finding that older adults perceive family and societal matters as more meaningful than younger adults, while younger adults derived more meaning from materialistic concerns than their older counterparts [14]. Although losses characterize old age, the findings show that there are probably differences in the intensity of the losses manifested throughout the entire aging period. Moreover, among the oldest age group (above 85 years) adverse life experiences may reduce a sense of coherence, a vital aspect of MIL [75]. The current results underscore are in line with the findings of notion an integrative literature review suggesting that maintaining MIL at older ages may be challenging [19]; in particular for non-religious older adults, therefore, it is essential to develop and implement intervention plans. On a clinical level, logotherapy, a type of psychotherapy based on the assumption that the search for meaning, even at times of suffering, includes a possible resolution to human suffering [76] may help older adults find personal MIL. Facilitating a combination of individual and community interventions may be essential to enriching MIL among older adults, such as promoting community contact and engagement including intergenerational volunteer-based neighborhood programs.
Added Reference:
- Frankl, V.E. The will to meaning: Foundations and applications of logotherapy. World Pub. Co 1984.
We also obtained a native English speaker to thoroughly review our paper and changes were made for clarification. Examples include:
** Although meaning in life is essential in all stages of life, the psychosocial development theory emphasizes the importance of MIL in old age was highlighted by the psychosocial development theory [9], postulating that in the eighth stage older adults are faced with the crisis of integrity vs. despair regarding whether their lives were meaningful.
** The current results underscore are in line with the findings of notion an integrative literature review suggesting that maintaining MIL at older ages may be challenging [19]; in particular for non-religious older adults, therefore, it is essential to develop and implement intervention plans.
Finally, we corrected the numbers of our citations in the text of the paper and in the References, in accordance with the additional articles that we have now added.
We once again thank Reviewer #4 for the valued comments that helped improve our paper.
*******************************************************
Reviewer 2 Report
The paper addresses different theoretical models of Meaning in Life (Meaning in Life MIL) in older adults and their relationship with the perception that people have of themselves. In this sense, the main point of the study is the analysis of demographic, psychosocial and personality factors, which affect the meaning of life in the elderly. In its current state, I consider that the paper has serious shortcomings that prevent its publication. The most relevant reasons are given below.
Introduction: it contains a rather plain account of the different variables to be measured, but the main reason for the study is not well explained: Why should older people differ from other populations on Meaning in Life? The answer is explained through different variables collected from other studies with younger adults. But there is no clear argumentation as to whether these are appropriate in young people.
Materials and methods. The information of the sample is very poor. No ranges are provided, and there seems to be no assessment of cognitive status or educational level. All these are crucial factors in the behavior of older adults. The statistical design is correct. However, I have doubts as to whether it is the most appropriate for the objectives of the study, perhaps other statisticians would provide more information.
One error: the data from the ethical committee approval file (line 55) do not appear.
Independent variables. They are defined by demographic variables and measurement instruments. It is not clear how the variable religiosity is defined, whether it means atheist versus any level of religiosity, or maybe differences in levels of religiosity. This would need an explanation. The locus of control scale has a very low alpha of .603. The same applies for the optimism scales (.672). This raises doubt on its reliability. It would be necessary to explain why this occurs and if it affects the conclusions. It may mean that it has measured poorly in older people.
The dependent variable is perhaps the most serious issue. It is measured by means of a 5-item scale. They do not present the items or how they were obtained. Nor any data that would allow us to evaluate the validity of the measurement with the construct proposed or, simply, that it is not a circular design.
Results. As expected, only demographic constructs are included in the first step. With all the variables in the analysis, 46.6% of the variance is explained. This indicates that the design is not well elaborated because there are variables that explain the construct that have not been considered. Other than that, the results are quite predictable and call into question the measurement of the dependent variable. MIL seems to depend on whether one has done well in life.
Discussion. Surprisingly, the main conclusion is that as the older adult have losses in the variables analyzed they give lower scores in the MIL. And vice versa. Then we should conclude that there are no differences with young people, if they also have losses, wouldn’t they obtain the same result? It seems that this aspect would be the most interesting for future studies. To compare whether the behavior of older adults is similar or different from that of younger ones.
The conclusion that states that younger elders perceive family issues as more significant than older ones doesn’t seem supported by the available data. At least not with the method used. There is a severe deficit in the interpretation of the results with conclusions that have not been obtained in the data. Another surprising conclusion is that "maintaining MIL at older ages is a challenge" (lin. 241). Besides, authors state that "it is common for people to gain MIL from religion and spirituality, especially as they age". Is it implied that causalities are obtained in a study that uses descriptive procedures? Given that it is declared that "it is essential to promote social programs within existing religious systems so that the contribution of religion to MIL can be further strengthened", it would seem so.
The rest of the discussion about personality factors is difficult to follow without further understanding of the measurement of the dependent variable.
Author Response
Reviewer #3: Comments and Suggestions for Authors
The paper addresses different theoretical models of Meaning in Life (Meaning in Life MIL) in older adults and their relationship with the perception that people have of themselves. In this sense, the main point of the study is the analysis of demographic, psychosocial and personality factors, which affect the meaning of life in the elderly. In its current state, I consider that the paper has serious shortcomings that prevent its publication. The most relevant reasons are given below.
Answer: We thank Reviewer #3 for the important feedback as well as for the helpful suggestions for revising and improving our manuscript. We have revised the paper in accordance.
Introduction: it contains a rather plain account of the different variables to be measured, but the main reason for the study is not well explained: Why should older people differ from other populations on Meaning in Life? The answer is explained through different variables collected from other studies with younger adults. But there is no clear argumentation as to whether these are appropriate in young people.
Answer: We thank the Reviewer for this vital comment, and for helping us clarify the importance of the study of MIL specifically among the older population. In order to expand the explanation of the importance and the purpose of the study, we have added the sections written in blue in the Introduction section:
Meaning in life is regarded as a personal experience [6] linked with psychological well-being throughout the lifespan [7]. Nevertheless, MIL varies during the lifespan, with some authors suggesting that life events and aging change a person's understanding of MIL [8], while others maintain that older age heightens one's sense of MIL [6]. Although meaning in life is essential in all stages of life, the psychosocial development theory emphasizes the importance of MIL in old age was highlighted by the psychosocial development theory [9], postulating that in the eighth stage older adults are faced with the crisis of integrity vs. despair regarding whether their lives were meaningful. Today, the final age stage is much longer and is divided into periods on a continuum from young older people (65+) the so-called "baby boomers" to older older people (80+) the "silent generation" [10]. Coping with the crisis of integrity vs. despair may change throughout the years of old age and may affect MIL in this population. Indeed, extensive research has underscored the significance of MIL in old age by linking MIL with diverse manifestations of successful aging [11] including better physical health, lower mortality [12], slower cognitive decline [13], higher subjective well-being [11], positive affect [13], fewer depressive symptoms [13], and better psychological adjustment [14]. Moreover, research suggests that MIL after traumatic experiences promotes post-traumatic growth [15], including older adults who survived early life trauma [16]. In an effort to explore factors contributing to variability in MIL in general, and particularly among older adults, previous studies explored single variables including age [17], culture [18], human relationships [19]and engagement in activities [20]. Nevertheless, conceptual models integrating diverse variables are needed in order to broaden the understanding of MIL [1]. In this light, fostering older adults' well-being requires a more profound knowledge of the factors that promote a sense of MIL in older age [21]. Therefore, the current research focuses on possible differences that may apply to this coping on their MIL. Identifying the factors that promote or diminish MIL in old age could help to develop appropriate therapeutic intervention methods and even adjust public policy. Hence, Specifically, the current study aims to present an integrative model of factors contributing to MIL among older adults, namely background characteristics (gender, age, employment status, religiosity), personality characteristics (locus of control, self-efficacy, optimism), and psycho-social factors (psychological distress and loneliness).
Added bibliography:
- Schnell, T. Sources of meaning and meaning in life questionnaire (SoMe): relations to demographics and well-being. J. Posit. Psychol. 2009 4, 483–499.
- King, L.A.; Hicks, J.A.; Krull, J.L.; Del Gaiso. A.K. Positive affect and the experience of meaning in life. J Pers Soc Psychol 2006 90(1) ,179-196.
- Reker, G.T.; Peacock, E.J.; Wong, P.T. Meaning and purpose in life and well-being: a life-span perspective. J Gerontol 1987 42, 44–49.
- Lissitsa, S.; Zychlinski, E.; Kagan, M. The Silent Generation vs Baby Boomers: Socio-demographic and psychological predictors of the “gray” digital inequalities. Comput Hum Behav 2022 128, 107098.
- MacKinlay, E. The Spiritual Dimension of Ageing. Jessica Kingsley Publishers. 2017.
Materials and methods. The information of the sample is very poor. No ranges are provided, and there seems to be no assessment of cognitive status or educational level. All these are crucial factors in the behavior of older adults.
Answer: Following the reviewer's comment, we expanded the information of the sample. For example, we added the following information regarding the level of formal education as well as adding ranges for all the characteristics of the sample by which they can be calculated (variables on the ratio scale):
Community dwelling Israeli adults aged 65 and older participated in the study (N = 751; 59.4% women). Their age ranged from 65 to 85 years (M = 72.27, SD = 6.28) and their formal education level ranged from 4 to 30 years (M = 13.78, SD = 3.35). Of the respondents, 59.4% were religious, 61.7% were unemployed, 65.9% were married or in a committed relationship, 21.2% widowed, 10.4% divorced, and 2.5% single. Most respondents (54.3%) classified themselves as having a medium socioeconomic status, 38.1% as having a high status, and 7.6% as having a low status.
The statistical design is correct. However, I have doubts as to whether it is the most appropriate for the objectives of the study, perhaps other statisticians would provide more information.
Answer: As the reviewer pointed out, the statistical model is indeed correct. The purpose of the study was to present an integrative model of factors contributing to MIL among older adults, namely background characteristics (gender, age, employment status, religiosity), personality characteristics (locus of control, self-efficacy, optimism), and psycho-social factors (psychological distress and loneliness).
In order to examine the unique contribution of the differentiated content worlds and not only of each variable separately, aimed to explain the dependent variable, a hierarchical regression was conducted. Replacing the word "step" instead of the word "model" may have created a certain ambiguity in the reading of the Table. Therefore, following the reviewer's justified comment, we changed the term to "step" in the Results section.
One error: the data from the ethical committee approval file (line 55) do not appear.
Answer: We had not added the data for sake of blinding the paper for review. We sent this information to the handing Editor on submission. We have now added this information as requested by the Reviewer as follows:
The study was approved as a non-clinical human research by the Ariel University Ethics Committee (approval number AU-SOC-EZ-20191029).
Independent variables. They are defined by demographic variables and measurement instruments. It is not clear how the variable religiosity is defined, whether it means atheist versus any level of religiosity, or maybe differences in levels of religiosity. This would need an explanation.
Answer: We asked the respondents to indicate whether they define themselves as (0-religious, 1-not religious). This information is now included in the Research instruments section.
The locus of control scale has a very low alpha of .603. The same applies for the optimism scales (.672). This raises doubt on its reliability. It would be necessary to explain why this occurs and if it affects the conclusions. It may mean that it has measured poorly in older people.
Answer: Indeed, in retrospect, the internal consistency reliability of the optimism and locus of control scales appeared to be lower than the customary threshold of 0.8. Hence, following the reviewer’s comment, we added this as a research limitation as follows:
Also, the internal consistency reliability of the optimism and locus of control scales in the current study is lacking. Therefore, future studies should examine whether these scales are indeed suitable for older people and whether they are sensitive to the cultural aspect of the population in which they were administered. In future studies the current research model should also be explored using alternative research tools.
The dependent variable is perhaps the most serious issue. It is measured by means of a 5-item scale. They do not present the items or how they were obtained. Nor any data that would allow us to evaluate the validity of the measurement with the construct proposed or, simply, that it is not a circular design.
Answer: Following the reviewer's comment, we moved Table 1 for clarity purposes and added the following information about the dependent variable:
Presence of meaning in life. The Meaning in Life Questionnaire (revised) [5] consists of two five-item factors designed to measure presence of meaning (to what degree respondents feel that their lives have meaning), and search for meaning (to what degree respondents strive to find meaning and understanding in their lives). The present study referred only to the presence of meaning in life subscale, which included items such as "I understand my life’s meaning", "I have a good sense of what makes my life meaningful". Items were rated on a 7-point Likert-type scale ranging from 1 (absolutely untrue) to 7 (absolutely true). The final index was calculated as the sum of responses to the five items, with scores ranging from 7 to 35 and higher scores indicating a higher level of presence of MIL. The current study’s internal consistency reliability (Cronbach's alpha) was .864.
Results. As expected, only demographic constructs are included in the first step. With all the variables in the analysis, 46.6% of the variance is explained. This indicates that the design is not well elaborated because there are variables that explain the construct that have not been considered. Other than that, the results are quite predictable and call into question the measurement of the dependent variable. MIL seems to depend on whether one has done well in life.
Answer: As evident from our literature review, MIL is a complex variable that might be explained by an extensive range of factors. Our study, similar to most studies on MIL and in other domains, does not presume to reach a full explanation of the dependent variable. It is doubtful whether this is possible on a statistical or theoretical level, considering that there are also latent variables of which we may lack knowledge. At the same time, we are aware that additional variables might explain MIL, and in this study, we propose additional research variables that should be considered in future studies. The current study joins previous research and contributes to the existing knowledge on MIL among older people. The findings show that, besides the significance of the psychosocial variables in explaining MIL, demographic variables might help identify older adults whose MIL is lower than others and provide them with proper attention and assistance. Moreover, the study reveals that individuals' personality traits significantly affect their perception of life as meaningful. This finding might help develop new directions for intervening and treating older people, as it stresses the connection between their al mental resources and their perception of meaning in life.
Discussion. Surprisingly, the main conclusion is that as the older adult have losses in the variables analyzed they give lower scores in the MIL. And vice versa. Then we should conclude that there are no differences with young people, if they also have losses, wouldn’t they obtain the same result? It seems that this aspect would be the most interesting for future studies. To compare whether the behavior of older adults is similar or different from that of younger ones.
Answer: We thank the Reviewer for this important point. The comparison of the behavior of older adults with that of younger ones is indeed an intriguing direction for future studies. We have added this important suggestion to in the Conclusion section as follows:
We suggest that future studies replicate a similar model to compare older adults' behavior with that of their younger counterparts.
The conclusion that states that younger elders perceive family issues as more significant than older ones doesn’t seem supported by the available data. At least not with the method used. There is a severe deficit in the interpretation of the results with conclusions that have not been obtained in the data.
Answer: We apologize for the confusion of this sentence and thank the Reviewer for pointing this out. We have made three amendments to clarify this.
Firstly, in the Introduction section, under the heading Background Characteristics and Meaning in Life we included the age groups that the researchers in previous work referred to, as follows:
MIL may relate to background characteristics, demonstrated in a study examining differences in age, gender, and ethnicity in relation to MIL. Findings show that younger participants (40 and below) perceive materialistic concerns as more significant to MIL than older participants (60 and above). In contrast, older participants reported family and communal values as more essential to MIL than younger participants [18].
Secondly, in the Introduction, we added the following in the section on Loneliness:
Later years create conditions whereby older people exhibit more loneliness than younger counterparts [64], with studies indicating about one-third of older people experiencing loneliness [65], and those aged 80 and over regularly feeling lonely [60]. It was also found that older men with the personality characteristics of agreeableness, emotional stability, and openness reported feeling less lonely [66]. Specifically, Rresearch also links loneliness with reduced MIL [67]; both are essential for daily functioning and maintaining mental health and well-being in older age [68].
Added Reference:
- Itzick, M.; Kagan, M.; Zychlinski, E. The big five personality traits as predictors of loneliness among older men in Israel. J Psychol 2020 154(1), 60-74.
Thirdly, in order to clarify that we referred to older adults at different stages of old age we rewrote the sentence in the Discussion section as follows:
The study’s findings reveal that younger participants (younger older adults) reported higher MIL levels than older participants (older older). This may be explained by losses associated with aging, including degenerating physical and mental health and decreasing intimate relationships [74]. Previous studies have highlighted the effect of age on MIL, finding that older adults perceive family and societal matters as more meaningful than younger adults, while younger adults derived more meaning from materialistic concerns than their older counterparts [14]. Although losses characterize old age, the findings show that there are probably differences in the intensity of the losses manifested throughout the entire aging period. Moreover, among the oldest age group (above 85 years) adverse life experiences may reduce a sense of coherence, a vital aspect of MIL [75]. The current results underscore are in line with the findings of notion an integrative literature review suggesting that maintaining MIL at older ages may be challenging [19]; in particular for non-religious older adults, therefore, it is essential to develop and implement intervention plans. On a clinical level, logotherapy, a type of psychotherapy based on the assumption that the search for meaning, even at times of suffering, includes a possible resolution to human suffering [76] may help older adults find personal MIL. Facilitating a combination of individual and community interventions may be essential to enriching MIL among older adults, such as promoting community contact and engagement including intergenerational volunteer-based neighborhood programs.
Added Reference:
- Frankl, V.E. The will to meaning: Foundations and applications of logotherapy. World Pub. Co 1984.
Another surprising conclusion is that "maintaining MIL at older ages is a challenge" (lin. 241). Besides, authors state that "it is common for people to gain MIL from religion and spirituality, especially as they age". Is it implied that causalities are obtained in a study that uses descriptive procedures? Given that it is declared that "it is essential to promote social programs within existing religious systems so that the contribution of religion to MIL can be further strengthened", it would seem so.
Answer: We thank the Reviewer for this important point. In order to clarify that our conclusions are in line with previous data, we wrote the sentence that "maintaining MIL at older ages is a challenge" as follows:
The current results underscore are in line with the findings of notion an integrative literature review suggesting that maintaining MIL at older ages may be challenging [19]; in particular for non-religious older adults, therefore, it is essential to develop and implement intervention plans.
Moreover, in order not to imply causalities we have written the sentence as follows:
Religious participants reported more MIL than non-religious participants, supporting previous studies indicating that it is common for people to gain MIL from religion and spirituality, especially as they age [27]. Our findings also support the notion that those who possess religiously-based perceptions of the world (e.g., regarding death) have a more profound sense of MIL [28]. Based on the current findings, together with the findings of both qualitative studies [27] and quantitative studies [26]. Ffrom a policy planning perspective, it is essential to it may be useful to promote social programs within existing religious systems so that the contribution of religion to MIL can be further strengthened.
The rest of the discussion about personality factors is difficult to follow without further understanding of the measurement of the dependent variable.
Answer: We thank the reviewer for this constructive comment and hope that following our clarification of the dependent variable this is now clear.
We also corrected the numbers of our citations in the text of the paper and in the References, in accordance with the additional articles that we have now added.
We once again thank Reviewer #3 for the valued comments that helped improve our paper.
*******************************************************
Reviewer 3 Report
The focus on the selected variables could be explained a little further. Mixed/unclear findings on the link between certain variables and MIL appeared to be the justification for the focus in some instances, but in others it was unclear. For example, it was noted that studies have not looked at the link between locus of control and MIL - what prompted the authors to focus on it?
Really good engagement with lots of relevant literature.
Logotherapy could be defined for those unfamiliar this the term.
Interesting set of well supported findings presented and their implications considered.
Author Response
Reviewer #2: Comments and Suggestions for Authors
Answer: We thank Reviewer #2 for the important feedback as well as for the helpful suggestions for revising and improving our manuscript. We have revised the paper in accordance.
The focus on the selected variables could be explained a little further. Mixed/unclear findings on the link between certain variables and MIL appeared to be the justification for the focus in some instances, but in others it was unclear. For example, it was noted that studies have not looked at the link between locus of control and MIL - what prompted the authors to focus on it?
Answer: Following the reviewer's comment, we added the following explanation:
Locus of control refers to the belief that events in life are controlled internally or externally [34]. People with an internal locus of control perceive self-accountability for shaping and controlling their lives, while those with an external locus of control perceive being controlled by luck, fate, opportunities, others, and events [35]. A high internal locus of control was linked with physical and mental outcomes including lower pain [36], better physical functionality [36], higher self-reported health [37], lower mortality [38], and better quality of life [39]. Meaning in life is based on feelings of coherence, purpose and existential mattering [40], although both of these concepts relate to feelings of coherence and purpose. However, there is a lacuna in studies examining the link between locus of control and MIL.
Added Reference:
- Costin, V.; Vignoles, V.L. What do people find most meaningful? How representations of the self and the world provide meaning in life. J Pers 2022 90, 541–558.
Really good engagement with lots of relevant literature.
Answer: We thank the Reviewer for the positive feedback
Logotherapy could be defined for those unfamiliar this the term.
Answer: We thank the Reviewer for the important point. We have added a definition of Logotherapy as follows:
On a clinical level, logotherapy, a type of psychotherapy based on the assumption that the search for meaning, even at times of suffering, includes a possible resolution to human suffering [76] may help older adults find personal MIL.
Added Reference:
- Frankl, V.E. The will to meaning: Foundations and applications of logotherapy. World Pub. Co 1984
Interesting set of well supported findings presented and their implications considered.
Answer: We thank the Reviewer for the positive feedback.
We also corrected the numbers of our citations in the text of the paper and in the References, in accordance with the additional articles that we have now added.
We once again thank Reviewer #2 for the valued comments that helped improve our paper.
*******************************************************
Reviewer 4 Report
Thank you for this interesting paper. With our aging population quality of life or meaning in life is an important outcome. The background was an excellent review of the literature. The methods were appropriate and reported well. The results were interesting and again well reported.
In you discussion you presented your findings and discussed how they related to the existing literature and speculated on why they might be the same or different. The conclusion covered your results.
Author Response
Reviewer #1: Comments and Suggestions for Authors
Thank you for this interesting paper. With our aging population quality of life or meaning in life is an important outcome. The background was an excellent review of the literature. The methods were appropriate and reported well. The results were interesting and again well reported.
In you discussion you presented your findings and discussed how they related to the existing literature and speculated on why they might be the same or different. The conclusion covered your results.
Answer: We thank Reviewer #1 for the positive feedback of our manuscript.

Round 2
Reviewer 1 Report
Dear
The authors faithfully revised the major requests.
Thank you, However, I ask for some more corrections
Author Response
Reviewer 1: The authors faithfully revised the major requests.
Thank you, However, I ask for some more corrections.
Answer: We thank the Reviewer for the helpful suggestions for revising and improving our manuscript. We have made more corrections as requested by the Reviewer as follows:
- we have now moved the characterization of the sample to the results section under the heading "characterization of the sample", as well as Table 1 with a brief description.
- To avoid confusion with the definitions of clinical studies (all studies in humans) and non-clinical studies (do not include humans), the following sentence “The study was approved as a non-clinical human research by the Ariel University Ethics Committee (approval number AU-SOC-EZ-20191029)” was changed to: “The study was approved by the Ariel University Ethics Committee (approval number AU-SOC-EZ-20191029)”.
- The name of the single author, Kovaleva, was added before the citation number as follows:
Locus of control was assessed by the Short Scale for the Assessment of Locus of Control (IE-4) designed by Kovaleva [69], including two items on the Internal Locus of Control scale and two items on the External Locus of Control scale.
- At the end of the discussion we changed to word "lacking" to "low" and cited references with Cronbach’s α scores as follows:
Also, the internal consistency reliability of the optimism and locus of control scales in the current study is lacking low, while in contrast to previous studies optimism had a strong internal reliability of 0.82 [81] and internal and external locus of control had an internal reliability of 0.69 and 0.71 respectively in a Norwegian sample and 0.80 and 0.60 respectively in a German-speaking sample [82].
Added References:
- Adebayo, D.F.; Akintola, A.; Olaseni, A. Adherence to COVID-19 preventive behaviours: the implication of life orientation and sociodemographic factors among residents in Nigeria. Psychol, 2022 13(4), 469-481.
- Krampe, H.; Danbolt, L.J.; Haver, A.; Stålsett, G.; Schnell, T. (2021). Locus of control moderates the association of COVID-19 stress and general mental distress: results of a Norwegian and a German-speaking cross-sectional survey. BMC Psychiatry 2022 21(1), 1-13.
We once again thank the Reviewer for the valued comments that helped improve our paper. We hope that it is now ready for acceptance.
*******************************************************